# Seroprevalence of diphtheria and tetanus antibodies among children and adolescents in high- and low-immunization coverage areas in the Lao People's Democratic Republic

Yuta Yokobori[1]*, Masaaki Iwaki[2,3], Miyuki Kimura[2], Noriko Kitamura[4], Haru Angelique Hoshino[5], Bandith Soumphonphakdy[6], Chansay Patthammavong[6], Bangone Tannavong[6], Amphai Khamsing[6], Phonethipsavanh Nouanthong[6], Mathida Thongseng[6], Masahiro Sano[1], Ayako Masu[1], Yuriko Egami[1], Eiichi Shimizu[1], Shinsuke Miyano[1], Kento Mitani[1], Sumiyo Okawa[1], Moe Moe Thandar[1], Hyun Kim[2], Masahiro Yutani[2], Mitsutoshi Senoh[2], Masahiko Hachiya[1]

**1** Bureau of Global Health Cooperation, Japan Institute for Health Security, Tokyo, Japan, **2** Department of Bacteriology II, National Institute of Infectious Diseases, Japan Institute for Health Security, Tokyo, Japan, **3** Department of Quality Control, Japan Institute for Health Security, Tokyo, Japan, **4** Center for Infectious Disease Epidemiology, National Institute of Infectious Diseases, Japan Institute for Health Security, Tokyo, Japan, **5** Department of Immunization Research, National Institute of Infectious Diseases, Japan Institute for Health Security, Tokyo, Japan, **6** Ministry of Health, Vientiane, Lao People's Democratic Republic

* yokobori.y@jihs.go.jp

## Abstract

### Introduction

Diphtheria and tetanus remain significant public health concerns in low- and middle-income countries (LMICs), including the Lao People's Democratic Republic (Lao PDR). Although national infant immunization coverage has improved, substantial subnational disparities persist, and no booster doses are administered beyond infancy. This study aimed to assess population-level immunity to diphtheria and tetanus among children and adolescents in high- and low-coverage areas of Lao PDR.

### Methods

A cross-sectional sero-epidemiological survey was conducted from September to October 2024 in Oudomxay (high coverage) and Xaisomboun (low coverage) provinces. A total of 960 participants aged 1–19 years were selected through multistage cluster sampling. Demographic and vaccination data were collected, and dried blood spots were tested for anti-diphtheria and anti-tetanus immunoglobulin G (IgG) using enzyme-linked immunosorbent assay. Seropositivity was defined as IgG level ≥0.1 IU/mL for diphtheria and ≥0.16 IU/mL for tetanus. Multilevel multivariate logistic regression identified factors associated with seropositivity.

**Data availability statement:** Due to ethical restrictions related to participant confidentiality, the individual-level data cannot be made publicly available. Aggregated, anonymized data underlying the findings of this study are available from the corresponding author and the affiliated institution, the Bureau of Global Health Cooperation, Japan Insite of Health Security (JIHS), https://kyokuhp.jihs.go.jp/eng/who-we-are/contact.html, upon reasonable request.

**Funding:** This study was supported by the National Center for Global Health and Medicine (NCGM) Intramural Research Fund (22A01 and 22K902C01), Japan, and the Grant for the National Immunization Program, Lao PDR (FY2024). The funders had no role in the conceptualization, design, data collection, analysis, decision to publish, or preparation of the manuscript.

**Competing interests:** The authors declare that they have no competing interests.

**Abbreviations:** aOR, adjusted odds ratio; DTP, diphtheria-tetanus-pertussis; ELISA, enzyme-linked immunosorbent assay; EPI, Expanded Program on Immunization; IgG, immunoglobin G; Lao PDR, Lao People's Democratic Republic; LMIC, low- and middle-income country; NIID, National Institute of Infectious Diseases; WHO, World Health Organization.

## Results

A total of 960 participants were enrolled; the response rate was 100%. Overall, 79.1% of participants reported receiving the three-dose pentavalent vaccine (Penta3), with coverage higher in Oudomxay (91.9%) than in Xaisomboun (66.3%). Seropositivity rate for both diphtheria and tetanus displayed a U-shaped age distribution, declining among school-aged children. Oudomxay showed higher seropositivity for both diseases. Multivariate analysis revealed lower seropositivity among participants aged 5–14 years and Hmong participants. Penta3 was significantly associated with tetanus, but not diphtheria, seropositivity.

## Conclusions

This study identified notable immunity gaps among school-aged children and ethnic minorities in Lao PDR. Findings support introducing WHO-recommended booster doses and implementing targeted strategies to enhance coverage equity and population-level protection.

## Introduction

Diphtheria and tetanus remain significant public health threats in many low- and middle-income countries (LMICs), including the Lao People's Democratic Republic (Lao PDR). Diphtheria, caused by toxigenic *Corynebacterium diphtheriae*, and tetanus, caused by *Clostridium tetani*, are potentially fatal if left untreated [1–3]. Although the Expanded Program on Immunization (EPI) has led to substantial global reductions in incidence, periodic diphtheria outbreaks continue, especially in Southeast Asia, where immunity gaps persist [4–6].

The World Health Organization (WHO) recommends three primary doses of the diphtheria-tetanus-pertussis (DTP) vaccine during infancy, followed by booster doses at ages 12–23 months, 4–7 years, and 9–15 years [7,8]. In Lao PDR, the national EPI schedule provides the pentavalent (Penta) vaccine (DTP–HepB–Hib) at 6, 10, and 14 weeks of age; however, routine booster doses beyond infancy have not been implemented [8,9]. While national three-dose pentavalent vaccine (Penta3) coverage has improved, considerable subnational variation remains. Some districts report high coverage, while others—especially those with remote or marginalized populations—continue to show low coverage [9–11]. Previous studies suggest that certain ethnic groups, such as the Hmong, may experience disproportionately lower vaccination uptake [11–14]. This highlights the importance of systematically monitoring population immunity levels to identify at-risk groups and preempt potential outbreaks. Inadequate surveillance of waning immunity may delay timely public health responses, exacerbating outbreak severity and spread.

Waning immunity following primary vaccination in infancy may contribute to diphtheria outbreaks, particularly among school-aged children and adolescents [6,15]. However, large-scale serological assessments of population-level immunity in Lao

PDR remain limited, hindering efforts to guide booster dose policies and targeted interventions. In this context, a sero-epidemiological study was conducted in both high- and low-coverage areas of Lao PDR to assess age-specific seroprevalence of diphtheria and tetanus antibodies and to identify vulnerable subpopulations who may benefit from targeted vaccination strategies. The objective was to generate robust, policy-relevant evidence to inform optimal immunization strategies for Lao PDR, including the introduction of WHO-recommended booster doses..

## Methods

### Study design and setting

A cross-sectional sero-epidemiological survey was conducted in the Lao People's Democratic Republic (Lao PDR). Four districts were purposively selected: Namor and Hoon districts in Oudomxay province, representing areas with high Penta3 coverage; and Anouvong and Longxan districts in Xaisomboun province, representing areas with low Penta3 coverage, according to Lao national immunization program data (S1 Table).

To situate external validity, Oudomxay generally exhibits routine immunization coverage above the national average, relatively better road connectivity, and a more mixed ethnic composition. By contrast, Xaisomboun has coverage below the national average, is mountainous, includes a higher proportion of Hmong communities, and faces greater access constraints. The two provinces were purposively selected to bracket high-coverage and low-coverage/remote contexts rather than to represent the national mean.

### Sampling strategy

Based on prior research experience in Laos [16,17], a design effect (sample variance/population variance) of 1.6 was assumed to calculate the sample size. As the seroprevalence of diphtheria and tetanus antibodies in the general pediatric population is unknown, an estimated prevalence of 50% (p = 0.5) was used, with a 95% confidence level (Z = 1.96), a 10% margin of error (d = 0.1), and an expected response rate of 80% (R = 0.8). These parameters yielded a target of 192 participants per age group.

$$n = 1.6 \times \frac{(1.96)^2 \times 0.5 \times (1 - 0.5)}{0.1 \times 0.1}/0.8 \approx 192$$

Participants were stratified into four age groups: 1–4 years, 5–9 years, 10–14 years, and 15–19 years. Thus, the minimum total under simple random sampling assumptions was 192 multiplied by 4 equal to 768. To operationalize the multistage cluster design and to ensure at least this minimum per stratum after potential non-response, an operational field target of 960 participants was set.

In each district, six villages were selected using probability proportional to size (PPS) sampling, yielding 24 villages [18]. Within each village, 10 participants per age group (40 per village) were randomly selected using household lists provided by local health authorities, for a total target of 24 multiplied by 40 equal to 960.

### Data collection

Trained surveyors conducted face-to-face interviews using structured questionnaires administered via Kobo Toolkit on tablets and paper from 23/09/2024 to 04/10/2024. Demographic characteristics, such as age group, gender, and ethnicity, self-reported vaccination history, and socioeconomic status (caregiver's education and household income) were collected. Ethnicity was self-identified; response options were Lao, Hmong, Khmer, and Other. For clarity, 'Hmong' refers to the Hmong ethnic group. When available, immunization cards were used to verify vaccination status; To address potential recall bias, vaccination status was verified using immunization cards when available, and standardized questionnaires were used by trained surveyors to enhance data accuracy.

## Specimen collection and laboratory analysis

Capillary blood was collected via finger prick onto Whatman 903® protein saver cards and air-dried at room temperature [19,20]. Dried blood spot samples were stored with desiccant and transported to the Department of Bacteriology II, National Institute of Infectious Diseases (NIID), Tokyo, Japan. After elution of serum fraction from the DBS [21], Anti-diphtheria and anti-tetanus immunoglobulin G (IgG) antibody titers were measured using a commercially available enzyme-linked immunosorbent assay (ELISA; IBL International GmBH, Germany), previously compared to the gold-standard toxin neutralization assay [21]. Prior to sample measurement, ELISA kits were calibrated against Vero cell neutralization assay (diphtheria) [22] and KPA agglutination kit [23] for assignment of unitage in International Units (IU) to samples.

The following cut-off values were used to define seropositivity: for diphtheria, IgG ≥ 0.1 IU/mL, as recommended by WHO for the minimum protective level [8,21,24,25]; and for tetanus, IgG ≥ 0.16 IU/mL, which is the lowest ELISA value reliably predictive of clinical protection according to WHO guidelines [26].

## Statistical analysis

Vaccination history (Penta3) was ascertained from caregiver recall and, when available, verified against immunization cards. When both sources were available and discrepant, the card record superseded recall. If the card was unavailable or indeterminate, the recall of participants was used; if neither source determined status, the vaccine history was coded as unknown. Among card holders, the percent agreements between recall and card records overall and by province and age were quantified.

Seropositivity rate of diphtheria and tetanus antibodies was estimated overall and by age group, province, and vaccination history, considered the sampling design and each individual's sampling weight. All estimates and standard errors were calculated by considering the multistage cluster sampling design and the weight of each sample. Chi-square ($\chi^2$) tests were used for categorical variables and Student's t-tests for continuous variables. Multilevel multivariate logistic regression models, accounting for village-level clustering, were used to identify factors associated with seropositivity, considering variations in vaccination coverage and social structure. Explanatory variables included age group, sex, vaccination status, province (high vs. low coverage), and ethnicity. Adjusted odds ratios (aORs) and 95% confidence intervals (CIs) were reported. Analyses were conducted using STATA version 16 (StataCorp, College Station, TX, USA). A p-value <0.05 was considered statistically significant.

Penta3 history was derived using a card-priority algorithm. For the regression analysis, 'vaccinated' was defined as card-documented Penta3 or caregiver recall of vaccination; 'not vaccinated' combined card-documented no doses, recall 'no', and 'I don't know/unknown' responses. This conservative rule avoids classifying undocumented or uncertain recall as vaccinated and aligns with programmatic coverage practices that do not count unknown status as vaccinated. For covariates with missing values (e.g., ethnicity, caregiver's education, household income, occupation), regression models used listwise deletion. We did not perform multiple imputation.

## Ethics approval and consent to participate

This study was approved by the Lao National Ethics Committee for Health Research, the National Center for Global Health and Medicine, Japan (NCGM-S-004865–00), the Institutional Review Board of the Ministry of Health, Lao PDR (2024.36), and the National Institute of Infectious Disease (NIID) (1819). Written informed consent was obtained from all participants or their legal guardians prior to enrollment.

## Result

A total of 960 participants were included, equally distributed between Oudomxay and Xaisomboun provinces. All sampled participants completed the survey (response rate 100%)." Table 1 presents the characteristics of the study population.

**Table 1. Characteristics of study participants in Oudomxay and Xaisomboun provinces.**

| Characteristics | Oudomxay (n = 480) | Xaisomboun (n = 480) | Total (n = 960) | P value |
|---|---|---|---|---|
| Penta 3 history (%) | | | | |
| Yes | 441 (91.9%) | 318 (66.3%) | 759 (79.1%) | 0.017 |
| No | 12 (2.5%) | 77 (16.0%) | 89 (9.3%) | |
| I don't know | 27 (5.6%) | 85 (17.7%) | 112 (11.7%) | |
| Sex (%) | | | | |
| Male | 185 (38.6%) | 216 (45.1%) | 401 (41.8%) | 0.263 |
| Female | 294 (61.4%) | 263 (54.9%) | 557 (58.0%) | |
| Ethnicity (%) | | | | |
| Lao | 188 (39.8%) | 99 (20.8%) | 287 (29.9%) | <0.001 |
| Khmer | 194 (41.1%) | 5 (1.1%) | 199 (20.7%) | |
| Hmong | 55 (11.7%) | 367 (76.9%) | 422 (44.0%) | |
| Others | 35 (7.4%) | 6 (1.3%) | 41 (4.3%) | |
| Caregiver's education (%) | | | | |
| No education | 56 (11.9%) | 71 (15.7%) | 127 (13.2%) | 0.013 |
| Primary | 157 (33.3%) | 86 (19.0%) | 243 (25.3%) | |
| Secondary | 105 (22.3%) | 98 (21.6%) | 203 (21.2%) | |
| High school | 123 (26.1%) | 85 (18.8%) | 208 (21.7%) | |
| University | 31 (6.6%) | 113 (24.9%) | 144 (15.0%) | |
| Household monthly income (million Kip) (%) | | | | |
| <2 | 107 (22.4%) | 54 (11.3%) | 161 (16.8%) | 0.095 |
| 2–4 | 191 (40.0%) | 223 (46.5%) | 414 (43.1%) | |
| 4–6 | 85 (17.8%) | 124 (25.8%) | 209 (21.8%) | |
| >6 | 58 (12.2%) | 75 (15.6%) | 133 (13.9%) | |
| Do not know | 36 (7.6%) | 4 (0.8%) | 40 (4.2%) | |
| Caregiver's occupation (%) | | | | |
| Public official | 56 (12.0%) | 108 (23.85) | 164 (17.1%) | 0.179 |
| Private sector | 5 (1.1%) | 12 (2.7%) | 17 (1.8%) | |
| Farmer/Fisherman | 244 (52.1%) | 149 (32.9%) | 393 (40.9%) | |
| Laborer | 44 (9.4%) | 30 (6.6%) | 74 (7.7%) | |
| Housewife | 85 (18.2%) | 109 (24.1%) | 194 (20.2%) | |
| Unemployed | 9 (1.9%) | 23 (5.1%) | 32 (3.3%) | |
| Others | 25 (5.3%) | 22 (4.9%) | 47 (4.9%) | |

Missing values: Sex [2], Ethnicity [11], Education [35], Income [3], Occupation [39].

The overall reported Penta3 coverage was 79.1%, significantly higher in Oudomxay (91.9%) than in Xaisomboun (66.3%) (P = 0.017). While the sex distribution was similar between provinces, ethnic composition differed substantially, with a greater proportion of Hmong ethnic group in Xaisomboun (76.9%) than in Oudomxay (11.7%) (P < 0.001). Caregiver education also varied: university-level education was more prevalent in Xaisomboun (24.9%), while primary education was more common in Oudomxay (33.3%) (P = 0.013). Differences in household income and caregiver occupation were observed but were not statistically significant.

Vaccination-card possession was 16.3% (156/960) overall (province-specific: Oudomxay 23.1%, Xaisomboun 9.4%; age-specific: 1–4 y 45.8%, 5–9 y 13.8%, 10–14 y 4.6%, 15–19 y 0.8%). Among card holders, percent agreement between recall and card for Penta3 was 92.3% overall, 96.4% in Oudomxay and 82.2% in Xaisomboun; by age: 91.8% (1–4

y), 90.9% (5–9 y), 100% (10–14 y), 100% (15–19 y; n small) (S2 Table). In addition, among participants without cards (n = 792), 604 (76.3%) had recall-only or unknown vaccination history.

Seropositive rates of diphtheria and tetanus antibodies differed significantly between provinces. The mean diphtheria seropositive rate was higher in Oudomxay (25.0 ± 3.44%) than in Xaisomboun (12.4 ± 2.31%; P = 0.017) (Fig 1). Similarly, tetanus seropositivity was greater in Oudomxay (65.6 ± 3.03%) compared to Xaisomboun (38.7 ± 2.33%; P = 0.005) (Fig 2).

Age-related seropositive patterns followed a U-shaped distribution for both diphtheria and tetanus: high among children aged 1–4 years, declining in those aged 5–9 years, and rising again in adolescents (Figs 2 and 3). This trend was more pronounced in Oudomxay; Xaisomboun showed consistently lower seropositivity across all age groups.

When stratified by vaccination history, participants with reported Penta 3 retained the U-shaped diphtheria seropositivity pattern, suggesting waning immunity followed by possible natural exposure at older ages. Unvaccinated individuals showed lower seropositivity in younger groups, with a steady increase by age, indicating gradual natural immunity

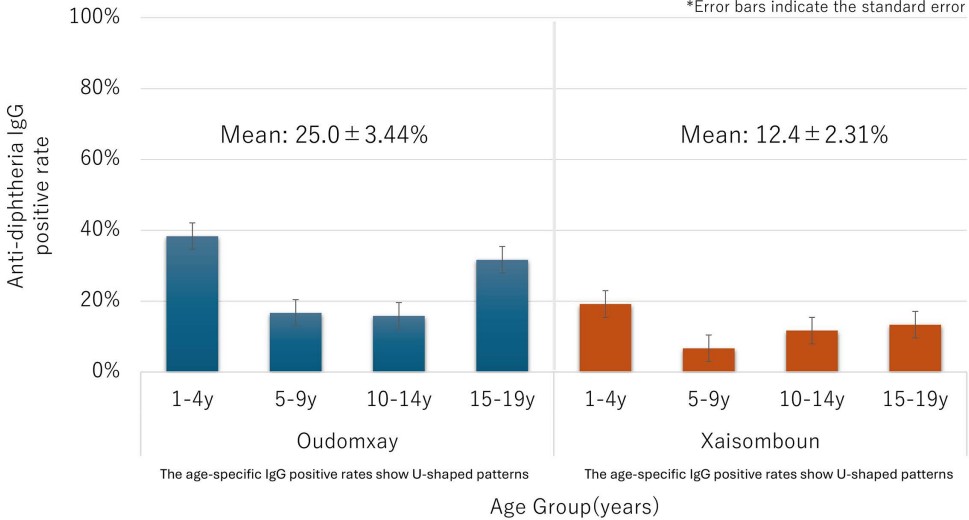

**Fig 1. Estimated anti-diphtheria IgG seropositive rate among different age groups.**

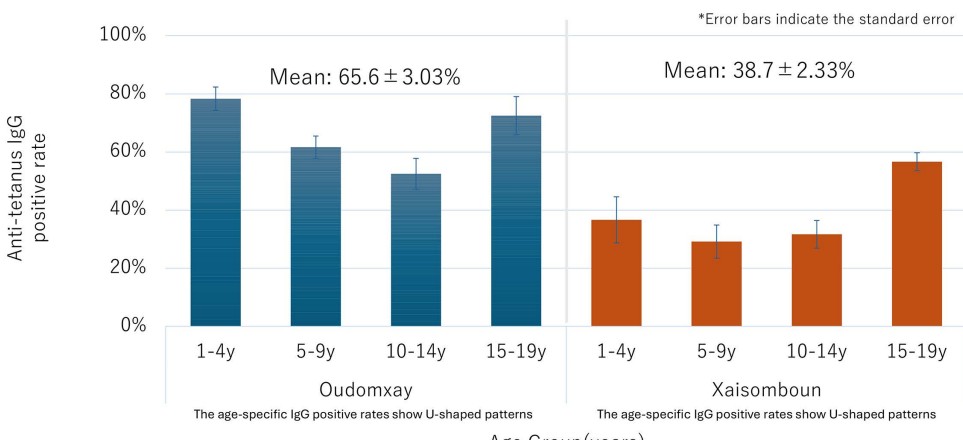

**Fig 2. Estimated anti-tetanus IgG seropositive rate among different age groups.**

acquisition (Fig 3). In the 1–4 y stratum, the 'no vaccination history' bar reflects a single recall-only case without a card (n = 1); this pattern most likely indicates recall error or undocumented vaccination, rather than persistence of maternal antibody. Tetanus seropositive rate among vaccinated participants was higher in younger age groups, declined during mid-childhood, and remained relatively stable in adolescence. Among unvaccinated individuals, tetanus seropositivity was generally lower at younger ages but gradually increased with age (Fig 4).

Multilevel multivariate logistic regression identified several factors associated with diphtheria and tetanus seropositivity (Table 2). For diphtheria, compared to children aged 1–4 years, seropositivity was significantly lower in those aged 5–9 years (aOR: 0.338, P = 0.008) and 10–14 years (aOR: 0.298, P = 0.022). No significant associations were found for sex, vaccination history, ethnicity, caregiver's education, or household income. For tetanus, Penta3 was associated with higher odds of seropositivity (aOR: 1.89, P = 0.049), while participants aged 10–14 years had lower odds (aOR: 0.35, P = 0.001) compared to the 1–4-year group. Hmong ethnicity was also associated with reduced odds of tetanus seropositivity (aOR: 0.33, P = 0.008).

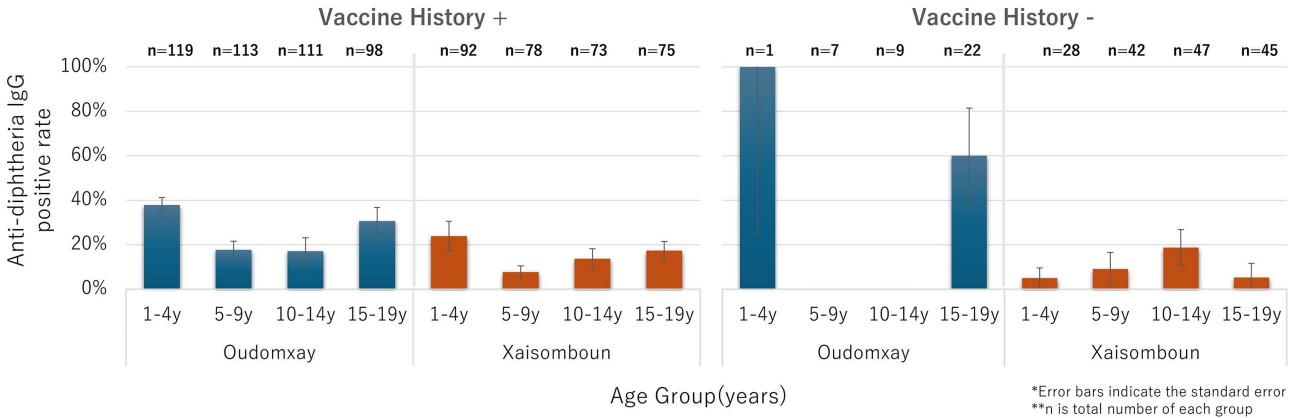

**Fig 3. Estimated anti-diphtheria IgG seropositive rates by Penta3 history and age group.**

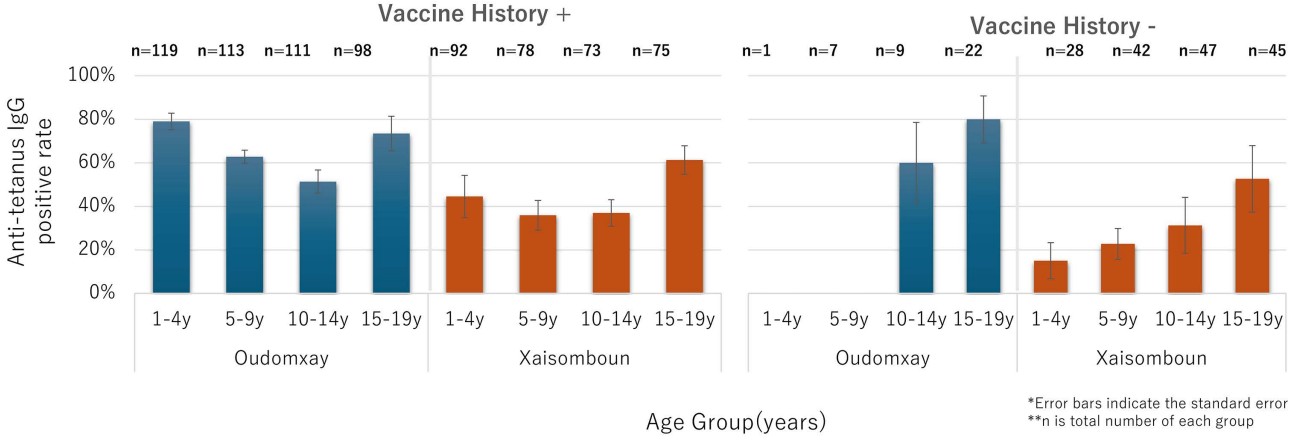

**Fig 4. Estimated anti-tetanus seropositive rates by Penta3 history and age group.**

**Table 2. Multilevel multivariate logistic regression analysis for seropositivity to diphtheria and tetanus.**

| Variables | Category | Diphtheria aOR* (95% CI) | P value | Tetanus aOR* (95% CI) | P value |
|---|---|---|---|---|---|
| Vaccine history | | | | | |
| | Vaccine history - | Ref | | Ref | |
| | Vaccine history + | 1.14 (0.53-2.49) | 0.667 | 1.89 (1.00-3.56) | 0.049 |
| Sex | | | | | |
| | Male | Ref | | Ref | |
| | Female | 0.96 (0.47-1.94) | 0.876 | 1.08 (0.64-1.82) | 0.730 |
| Age group | | | | | |
| | 1–4 y | Ref | | Ref | |
| | 5–9 y | 0.34 (0.18-0.65) | 0.008 | 0.56 (0.26-1.20) | 0.107 |
| | 10–14 y | 0.30 (0.12-0.77) | 0.022 | 0.35(0.23-0.54) | 0.001 |
| | 15–19 y | 0.63 (0.37-1.06) | 0.071 | 1.00 (0.53-1.88) | 0.993 |
| Ethnicity | | | | | |
| | Lao | Ref | | Ref | |
| | Khmer | 1.17 (0.43-3.20) | 0.705 | 0.85(0.46-1.56) | 0.526 |
| | Hmong | 0.55 (0.26-1.15) | 0.091 | 0.33(0.17-0.65) | 0.008 |
| | Others | 0.53 (0.12-2.32) | 0.323 | 0.89(0.49-1.63) | 0.647 |
| Caregivers' education | | | | | |
| | No education | Ref | | Ref | |
| | Primary | 1.05 (0.48-2.31) | 0.882 | 1.80 (0.81-3.98) | 0.117 |
| | Secondary | 1.24 (0.55-2.79) | 0.520 | 1.64 (0.59-4.58) | 0.270 |
| | High school | 0.82 (0.18-3.69) | 0.743 | 1.65 (0.87-3.15) | 0.099 |
| | University | 0.53 (0.16-1.74) | 0.229 | 0.92 (0.27-3.15) | 0.865 |
| Monthly household income | | | | | |
| | <2M LAK | Ref | | Ref | |
| | 2–4M LAK | 0.64 (0.37-1.12) | 0.098 | 1.00 (0.48-2.09) | 0.996 |
| | 4–6M LAK | 0.76 (0.38-1.52) | 0.354 | 1.17 (0.65-2.10) | 0.519 |
| | >6M LAK | 0.69 (0.22-2.08) | 0.421 | 1.44 (0.73-2.88) | 0.226 |
| | Do not know | 0.90 (0.44-1.85) | 0.728 | 0.46 (0.20-1.07) | 0.063 |

*Adjusted odds ratios (aOR) were estimated using multilevel multivariate logistic regression, with random intercepts for district and village to account for clustering. Models adjusted for vaccination status, age group, sex, ethnicity, caregiver's education, and household income.

## Discussion

This study offers critical insights into population-level immunity to diphtheria and tetanus among children and adolescents in Lao PDR, comparing provinces with differing vaccine coverage. Several key findings have implications for national immunization policy. First, both diphtheria and tetanus seropositivity rate exhibited a U-shaped distribution by age: high among children aged 1–4 years who had recently completed the primary Penta series, declining in those aged 5–9 years, and rising again among adolescents. The initial decline reflects well-documented waning of vaccine-induced antibodies in the absence of booster doses [6,8,20]. The subsequent increase in diphtheria seropositivity likely results from cumulative natural exposure to *Corynebacterium diphtheriae* [27], supported by recurring outbreaks in nearby Southeast Asian countries, including Vietnam [5,28,29]. These exposures may act as natural immunological boosters. In such an epidemiologic context, naturally boosted antibodies in both vaccinated and unvaccinated individuals can attenuate the association

between reported vaccination history and diphtheria seropositivity. In contrast, tetanus—where subclinical infection is rare and does not typically confer protective immunity [30]— remains more closely aligned with vaccination history; observed patterns likely reflect undocumented participation in Supplementary Immunization Activities (SIAs) or maternal immunization programs [11,12].

Second, significant provincial differences in seropositivity aligned with vaccination coverage. Government data show Penta3 coverage in Oudomxay province ranged from 98.3% to 110.3% over the past 5 years, compared to 46.1%–73.8% in Xaisomboun province. Correspondingly, both diphtheria and tetanus seropositivity were higher in Oudomxay. Structural and demographic factors—such as better health service access, improved cold chain management, and differences in ethnic composition [11,14] – may also contribute. The predominance of the Hmong ethnic group in Xaisomboun may reflect underlying structural and sociocultural determinants of vaccination uptake (e.g., geographic remoteness, transport and service availability, language/communication barriers, trust in public services) rather than a direct effect of ethnicity per se, consistent with evidence from rural Lao PDR that trust-building and tailored communication improve uptake among ethnic minority communities [14,31]. These findings underscore the need for geographically tailored vaccination strategies that address specific barriers faced by ethnic and underserved populations.

Third, stratified analysis by vaccination history revealed that vaccinated individuals retained the U-shaped seropositivity pattern, whereas unvaccinated individuals—particularly in Oudomxay—showed a steady increase with age, especially for diphtheria. This suggests ongoing community exposure to circulating diphtheria bacteria. In Xaisomboun, unexpectedly high seropositivity in some unvaccinated individuals, sometimes exceeding that of vaccinated peers, may reflect localized outbreaks, undocumented immunization efforts, recall bias in caregiver reporting, or small sample size limitations [5,11,32,33].

Fourth, multivariate analysis confirmed significantly lower odds of seropositivity among school-aged children (5–14 years), reinforcing the need for booster doses to sustain immunity beyond infancy [6,8]. Penta3 was strongly associated with tetanus, but not diphtheria, seropositivity—consistent with findings that tetanus toxoid induces more durable immune responses [34,35]. Lower seropositivity among Hmong participants further highlights the need for targeted strategies for underserved ethnic minorities [14,18], Such strategies may include culturally tailored education, materials in local languages, involvement of community leaders, and deployment of multilingual health workers [36,37].

These findings underscore the need for policy adaptation in Lao PDR. The introduction of WHO-recommended booster doses of diphtheria-tetanus-containing vaccines at school-entry age (4–7 years) is essential to counter waning immunity and reduce outbreak risk [8]. Efforts must also focus on enhancing outreach to underserved populations and improving data quality to strengthen equity and immunization program performance.

This study has several limitations. First, vaccination history for a large subset relied on caregiver recall, which may lead to non-differential misclassification [11,12]. However, the use of vaccination cards together with standardized data collection protocols helped minimize recall error [38]. Overall recall–card agreement was high (S2 Table), supporting the use of caregiver recall in this setting; nonetheless, among participants who recalled non-vaccination, 4 of 6 (66.7%) had vaccination documented on cards, indicating a modest underestimation of vaccination history when reliance is placed on recall alone. Rare strata with recall-only, no-card classification (e.g., 1–4 y 'no vaccination history' in Oudomxay, n = 1) are particularly vulnerable to recall bias, which tends to underestimate vaccination history and attenuate associations. Moreover, the primary exposure definition for Penta3 was conservative: only card-documented doses or an unequivocal recall of vaccination were classified as vaccinated, whereas 'I don't know/ unknown' and recall 'no' were grouped with not vaccinated. This rule reduces the risk of falsely classifying unverified recall as vaccinated, but may underestimate true vaccination coverage and bias associations toward the null. Second, although antibody levels were measured via validated ELISA, toxin neutralization assays remain the gold standard for assessing functional immunity [18]. Nevertheless, the ELISA used in this study has shown a strong correlation with the gold standard, supporting the reliability of sero-protection estimates [21]. Third, as subclinical tetanus infection is rare

and does not produce protective immunity, increased antibody levels among unvaccinated individuals likely reflect unrecorded vaccine exposure. In Lao PDR, SIAs are not documented on vaccination cards, which may explain these findings. This aligns with the known pathophysiology of tetanus and previous studies [2,23]. Fourth, regression models used listwise deletion for covariates with missing data; no imputation was performed. This approach assumes that missingness is ignorable conditional on included covariates; if violated, estimates may be biased—typically toward the null. Analytic sample sizes for each model are reported in Table 2. Since the proportion missing per covariate was low, the impact on inference is likely limited. Finally, while the selected provinces represent contrasting coverage contexts, the findings—such as the estimated seropositivity rate—may not be generalizable nationwide because the provinces and districts were purposefully selected based on National Immunization Program data. Still, inclusion of both high- and low-coverage areas enhances the study's relevance for guiding national policy and addressing immunity gaps across subpopulations [11]. With a 100% response rate, selection bias due to non-response is unlikely. Nonetheless, potential coverage/selection bias may persist due to the household listing and the purposive selection of provinces, which may affect generalizability beyond the sampled contexts.

## Conclusions

This study identified significant immunity gaps against diphtheria and tetanus among children and adolescents in Lao PDR, particularly in areas with low vaccine coverage and among ethnic minorities. The observed U-shaped seropositivity patterns reflect waning immunity after primary vaccination and possible natural exposure to diphtheria. These results support the introduction of WHO-recommended booster doses at school-entry age and emphasize the need for targeted interventions to improve vaccination coverage among underserved populations. Moreover, Policymakers should consider geographically differentiated strategies that prioritize vulnerable groups to strengthen immunization programs and overall public health outcomes.

## Supporting information

**S1 Table. The trend of Penta3 vaccination coverage over the past 10 years in each target district.**
(DOCX)

**S2 Table. Vaccination card possession and percent agreement in vaccination history between card records and recall.**
(DOCX)

## Acknowledgments

The authors acknowledge the contribution of the surveyors from the national, province, and district health offices in Lao PDR who are not coauthors of this article and all the participants in this survey.

## Author contributions

**Conceptualization:** Yuta Yokobori, Masaaki Iwaki, Noriko Kitamura, Bandith Soumphonphakdy, Chansay Patthammavong, Bangone Tannavong, Amphai Khamsing, Phonethipsavanh Nouanthong, Mathida Thongseng, Shinsuke Miyano, Masahiko Hachiya.

**Data curation:** Sumiyo Okawa, Moe Moe Thandar.

**Formal analysis:** Masaaki Iwaki, Miyuki Kimura, Noriko Kitamura, Haru Angelique Hoshino, Sumiyo Okawa, Moe Moe Thandar, Hyun Kim, Masahiro Yutani, Mitsutoshi Senoh.

**Funding acquisition:** Masahiko Hachiya.

**Investigation:** Yuta Yokobori, Bandith Soumphonphakdy, Chansay Patthammavong, Bangone Tannavong, Amphai Khamsing, Phonethipsavanh Nouanthong, Mathida Thongseng, Masahiro Sano, Ayako Masu, Yuriko Egami, Eiichi Shimizu, Shinsuke Miyano, Kento Mitani, Masahiko Hachiya.

**Methodology:** Yuta Yokobori, Bandith Soumphonphakdy, Chansay Patthammavong, Shinsuke Miyano, Masahiko Hachiya.

**Project administration:** Yuta Yokobori, Masahiko Hachiya.

**Resources:** Shinsuke Miyano, Masahiko Hachiya.

**Supervision:** Yuta Yokobori, Shinsuke Miyano, Mitsutoshi Senoh, Masahiko Hachiya.

**Validation:** Yuta Yokobori, Masahiko Hachiya.

**Visualization:** Masahiko Hachiya.

**Writing – original draft:** Yuta Yokobori.

**Writing – review & editing:** Masaaki Iwaki, Noriko Kitamura, Haru Angelique Hoshino, Bandith Soumphonphakdy, Chansay Patthammavong, Bangone Tannavong, Amphai Khamsing, Phonethipsavanh Nouanthong, Mathida Thongseng, Masahiro Sano, Ayako Masu, Yuriko Egami, Eiichi Shimizu, Shinsuke Miyano, Kento Mitani, Sumiyo Okawa, Moe Moe Thandar, Hyun Kim, Mitsutoshi Senoh, Masahiko Hachiya.

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
