## [Decision Letter · Decision Letter 0]

3 Oct 2025

Dear Dr. Yokobori,

plosone@plos.org . A rebuttal letter that responds to each point raised by the academic editor and reviewer(s). You should upload this letter as a separate file labeled 'Response to Reviewers'.

We look forward to receiving your revised manuscript.

Kind regards,

Ray Borrow, Ph.D., FRCPath

Academic Editor

PLOS ONE

Journal Requirements:

3. Please expand the acronym “NCGM” (as indicated in your financial disclosure) so that it states the name of your funders in full.

5. In this instance it seems there may be acceptable restrictions in place that prevent the public sharing of your minimal data. However, in line with our goal of ensuring long-term data availability to all interested researchers, PLOS’ Data Policy states that authors cannot be the sole named individuals responsible for ensuring data access (http://journals.plos.org/plosone/s/data-availability#loc-acceptable-data-sharing-methods).

6. Please amend the manuscript submission data (via Edit Submission) to include author Mistutoshi Senoh

7. Please amend your authorship list in your manuscript file to include author Mitsutoshi Senoh

8. We note that Figure 1 in your submission contain map images which may be copyrighted. All PLOS content is published under the Creative Commons Attribution License (CC BY 4.0), which means that the manuscript, images, and Supporting Information files will be freely available online, and any third party is permitted to access, download, copy, distribute, and use these materials in any way, even commercially, with proper attribution. For these reasons, we cannot publish previously copyrighted maps or satellite images created using proprietary data, such as Google software (Google Maps, Street View, and Earth). For more information, see our copyright guidelines: http://journals.plos.org/plosone/s/licenses-and-copyright.

Reviewers' comments:

Reviewer's Responses to Questions

**Comments to the Author**

1. Is the manuscript technically sound, and do the data support the conclusions?

Reviewer #1: Yes

Reviewer #2: Yes

2. Has the statistical analysis been performed appropriately and rigorously?

Reviewer #1: Yes

Reviewer #2: Yes

3. Have the authors made all data underlying the findings in their manuscript fully available?

Reviewer #1: Yes

Reviewer #2: Yes

4. Is the manuscript presented in an intelligible fashion and written in standard English?

Reviewer #1: Yes

Reviewer #2: Yes

Reviewer #1: Manuscript Title: Seroprevalence of Diphtheria and Tetanus Antibodies among Children and Adolescents in High- and Low-Immunization Coverage Areas in the Lao People's Democratic Republic

Manuscript Number: PONE-D-25-45990

Recommendation: Minor Revision

Summary Evaluation

This manuscript presents a well-designed cross-sectional sero-epidemiological study assessing immunity to Diphtheria and Tetanus among children and adolescents aged 1–19 years in Lao People's Democratic Republic, in Oudomxay (high coverage) and Xaisomboun (low coverage). Using multistage cluster sampling and validated ELISA assays, it provides robust data on age- and location-specific seroprevalence, revealing important immunity gaps in school-aged children and among ethnic minority groups, notably the Hmong.

The study is timely and policy-relevant. It highlights the risk posed by waning immunity in the absence of booster doses beyond infancy, supporting the introduction of World Health Organization-recommended boosters in Lao PDR.

However, several methodological clarifications are needed to strengthen transparency and interpretation. Most importantly, vaccination history should be more rigorously reported: the proportion of participants with vaccination cards (by age and province), recall–card concordance, and how discrepancies were handled. Missing data handling should be described, and 95% confidence intervals added to adjusted odds ratios (Table 2). The unexpected 100% Diphtheria seropositivity among 1–4 year-olds with “no vaccination history” likely reflects undocumented doses and should be re-examined with denominators and CIs provided. The sample size calculation and selection process should be clarified, and the representativeness of the two provinces briefly discussed. Finally, the discussion should frame lower Tetanus seropositivity among Hmong participants as likely driven by structural or geographic factors rather than ethnicity per se.

Overall, this is a timely and valuable study that would benefit from these clarifications.

Major comment :

1. Vaccination history: card availability and recall accuracy

Vaccination history was partly based on caregiver recall, but the frequency of card availability and the accuracy of recall are not reported.

- Please report the proportion of participants with vaccination cards (overall, by age and province) and assess concordance between recall and cards (e.g. agreement or κ).

- Clarify how discrepancies were handled and discuss potential misclassification bias in the Limitations.

- Consider presenting the results for three-level vaccine history (documented vaccinated / documented unvaccinated / recall-only or unknown).

2. Table 2: add 95% CIs

Table 2 shows adjusted ORs without confidence intervals.

- Please add the 95% CIs for all adjusted ORs and note in the footnote how clustering and covariates were handled.

3. Figure 4: 100% diphtheria seropositivity in 1–4 y with no vaccination history

It is unexpected that 1–4 y children with “no vaccination history” show 100% diphtheria seropositivity, as maternal antibodies wane by 12 months.

- Provide denominators (n) and 95% CIs for each bar.

- Recheck vaccination classification—acknowledge that very likely reflect undocumented doses.

- Consider separating “no doses” from “unknown/no card” and note that maternal IgG cannot explain this pattern.

4. Handling of Missing Data

Table 1 indicates missing values for key variables (e.g., Penta3 history, ethnicity, income). Please clarify how these were handled in the regression models (e.g., listwise deletion vs. imputation) as it can meaningfully affect regression estimates and interpretation of associations.

Minor Comments

1. Clarify Ethnicity Terminology

For clarity, explicitly state early in the Methods that Hmong is an ethnic group, to ensure international readers understand this categorization.

2. Interpretation of Ethnicity vs. Socioeconomic and Geographical Factors

The discussion attributes lower tetanus seropositivity to “Hmong ethnicity.” However, ethnicity may act as a proxy for geographic remoteness, structural exclusion, or unmeasured socioeconomic barriers (e.g., lower access to health services, transportation, or education). The authors note that differences in caregiver education, occupation, and income were not statistically significant, yet these contextual factors likely influence vaccination uptake. Consider reframing to emphasize that “ethnicity” may reflect underlying structural determinants and avoid implying a direct causal effect.

3. Association Between Vaccination and Seropositivity

Clarify that the lack of a significant association between vaccination history and diphtheria seropositivity likely reflects the persistent circulation of Diphtheria and frequent outbreaks affecting both vaccinated and unvaccinated individuals. This context is key to understanding why vaccination status was only associated with Tetanus seropositivity.

4. Sample Size Description

There is some ambiguity in the description of the sample size calculation and selection. The Methods state that “192 participants per group were selected, totaling 800,” yet also mention that “to support multistage cluster sampling, a target of approximately 960 participants was set.” This appears inconsistent and could confuse readers about how the final sample size was determined.

Please clarify:

- Precisely how the figure of 192 participants per age group was calculated (e.g., based on assumed prevalence, confidence level, design effect, expected response rate);

- why this initial total (192 × 4 age groups = 768 or “800” as written) differs from the final target of 960;

- and how this target was operationalized within the multistage cluster design (e.g., number of clusters/villages, participants per cluster, and any oversampling to account for non-response).

Providing a concise flow (calculation → adjustment for cluster design → final enrolled number) would help readers understand the sampling framework and ensure transparency of the study design.

5. Representativeness of Selected Provinces

Please clarify how the two selected provinces (Oudomxay and Xaisomboun) compare to national averages in terms of immunization coverage, demographics, and health service access. This would help readers assess the external validity and generalizability of the findings, given that the provinces were purposively chosen to contrast coverage levels.

6. Typographical Errors

- In the Statistical Analysis section, correct “seropositivity rare” to “seropositivity rate.”

- Consider

Reviewer #2: The study addresses an important public health issue by evaluating vaccination coverage and seroprevalence of diphtheria and tetanus among subjects aged 1-19 years, in Oudomxay and Xaisomboun provinces of Lao PDR. The use of serological data in combination with vaccination history provides valuable insight into population-level immunity and possible gaps in immunization coverage.

The manuscript is concise and presents significant results that could have meaningful implications for immunization strategies. However, several areas require clarification and expansion before the manuscript is acceptable for publication.

1. Please provide the response rate of the study in both the abstract and results sections. In addition, please discuss the potential implications of selection bias arising from non-response.

2. In Table 2, please provide 95% confidence intervals (CIs) alongside the adjusted odds ratios (aORs).

3. I recommend adding a graphical representation of the U-shaped seropositivity pattern by age group for each antigen (diphtheria and tetanus) stratified by province.

**Do you want your identity to be public for this peer review?** For information about this choice, including consent withdrawal, please see our Privacy Policy

Reviewer #1: No

Reviewer #2: No

---

## [Author Response · Author response to Decision Letter 1]

31 Oct 2025

27th Oct 2025

The Editors

PLOS ONE

Manuscript ID: PONE-D-25-45990

Manuscript title: Seroprevalence of Diphtheria and Tetanus Antibodies among Children and Adolescents in High- and Low-Immunization Coverage Areas in the Lao People’s Democratic Republic

Dear Editor:

I am writing to you in relation to the manuscript titled ‘Seroprevalence of Diphtheria and Tetanus Antibodies among Children and Adolescents in High- and Low-Immunization Coverage Areas in the Lao People’s Democratic Republic’ which was submitted to PLOS ONE by Yuta Yokobori and co-authors for consideration for publication. Following its initial review, we were asked to amend the manuscript in light of the reviewers’ comments. We have carefully reviewed the comments and have revised the manuscript accordingly. Our responses are given in a point-by-point manner below. Changes to the manuscript are shown by track change and highlighted by green

We hope that the revised manuscript is now suitable for publication. We look forward to hearing from you in due course.

Sincerely,

Dr. Yuta Yokobori

Bureau of International Health Cooperation

National Center for Global Health and Medicine (NCGM)

1-21-1 Toyama, Shinjuku-ku, Tokyo 162-8655, Japan

Tel: +81-3-3202-7181

Fax: +81-3-3202-7860

Email: y-yokobori@it.ncgm.go.jp

Responses to Reviewers(s)

We appreciate the reviewer’s thoughtful and constructive feedback on our manuscript. We sincerely appreciate the time and effort the reviewer has invested in reviewing our work. I made some revisions according to the comments and highlighted the parts by yellow. In response to the comments, we have made the following revisions to the manuscript.

Reviewer 1

Major comments

1. Vaccination history: card availability and recall accuracy

Vaccination history was partly based on caregiver recall, but the frequency of card availability and the accuracy of recall are not reported.

- Please report the proportion of participants with vaccination cards (overall, by age and province) and assess concordance between recall and cards (e.g. agreement or κ).

- Clarify how discrepancies were handled and discuss potential misclassification bias in the Limitations.

- Consider presenting the results for three-level vaccine history (documented vaccinated / documented unvaccinated / recall-only or unknown).

Response:

Thank you for the helpful suggestions. S2 Table to summarize the report availability and percent agreement (recall vs card) was newly created. The manuscript has been revised as follows:

• Card availability reported (S2 Table) in “Result”, Line 190-195 : overall 16.3% (156/960); by province—Oudomxay 23.1%, Xaisomboun 9.4%; by age—1–4 y 45.8%, 5–9 y 13.8%, 10–14 y 4.6%, 15–19 y 0.8%.

• Concordance (recall vs card) in “Result”, Line 190-195: Percent agreement is presented in S2 Table.

• Discrepancy rule in “Methods”, Line 142-147 : when both sources existed and disagreed, card superseded recall; if the card was absent/indeterminate, recall was used; if neither determined status, unknown was coded for descriptive summaries.

• Three-level history in “Result”, Line 190-195: a separate table was not added because documented vaccinated / documented unvaccinated are already shown in S2 Table. Instead, Results now state that among participants without cards (n = 792), 604 (76.3%) had recall-only or unknown vaccination history.

• Misclassification in “Limitation”, Line 291-301: while overall agreement was high, among those who recalled “non-vaccination,” 4/6 (66.7%) had vaccination documented on cards, indicating slight underestimation of vaccination history when relying on recall.

These changes address the request to quantify card availability, assess concordance, specify handling of discrepancies, discuss misclassification, and consider the three-level presentation without adding an additional table.

2. Table 2: add 95% CIs

Table 2 shows adjusted ORs without confidence intervals.

- Please add the 95% CIs for all adjusted ORs and note in the footnote how clustering and covariates were handled.

Response:

Thank you for the suggestion. We revised Table 2 as follows:

• The effect-size column now reads “aOR (95% CI)” and reports 95% confidence intervals for all adjusted ORs

• A table footnote was added to document clustering and covariate adjustment:

“Adjusted odds ratios (aOR) were estimated using multilevel multivariate logistic regression, with random intercepts for district and village to account for clustering. Models adjusted for vaccination status, age group, sex, ethnicity, caregiver’s education, and household income.”

3. Figure 4: 100% diphtheria seropositivity in 1–4 y with no vaccination history

It is unexpected that 1–4 y children with “no vaccination history” show 100% diphtheria seropositivity, as maternal antibodies wane by 12 months.

- Provide denominators (n) and 95% CIs for each bar.

- Recheck vaccination classification—acknowledge that very likely reflect undocumented doses.

- Consider separating “no doses” from “unknown/no card” and note that maternal IgG cannot explain this pattern.

Response:

Thank you for the suggestions. We revised Figure 4 (now Figure 3) and the text as follows:

• Denominators and 95% CIs added: Each bar now displays n and 95% CIs. In the 1–4 y “no vaccination history” stratum, n=1; the observed 100% corresponds to a 95% CI of 25–100.0%, indicating extreme imprecision.

• Rechecked classification & interpretation: The 1–4 y “no vaccination history” observation is just 1 case with a recall-only and no-card. This very likely reflects undocumented doses or recall error; maternal antibodies cannot plausibly explain seropositivity beyond 12 months.

• On separating “no doses” vs “unknown/no card”: Not implemented for 1–4 y because the relevant cell has n=1, making a split not feasible.

• We inserted relevant sentence in “Result”

4. Handling of Missing Data

Table 1 indicates missing values for key variables (e.g., Penta3 history, ethnicity, income). Please clarify how these were handled in the regression models (e.g., listwise deletion vs. imputation) as it can meaningfully affect regression estimates and interpretation of associations.

Response:

Thank you. Penta3 was handled separately by a conservative exposure rule: “vaccinated” only when card-documented or clearly recalled ‘yes’; “not vaccinated” combined card/recall ‘no’ and ‘I don’t know/unknown’. We conducted complete case analysis, and no imputation was used for Penta3. Regarding the missing data of other covariates, we used listwise deletion for covariates with missing data (e.g., ethnicity, caregiver’s education, income, occupation). We added the relevant sentences in “Method” , Line 164-166. We did not perform imputation for three reasons:

1. Low missingness: the proportion missing per covariate was small (e.g., ethnicity 11/960≃1.1%, education 35/960≃3.6%, income 3/960≃0.3%, occupation 39/960≃4.1%), so the loss of power was minimal and any gain from imputation would be limited.

2. Model validity: imputation would require cluster-aware, multinomial models for categorical covariates under unverifiable MAR assumptions; misspecification risks introducing bias larger than the gain from imputation.

3. Transparency & reproducibility: complete-case estimates with analytic number reported in Table 2 are straightforward to interpret and reproduce; we also acknowledge potential bias from listwise deletion in the Limitations.

Minor Comments

1. Clarify Ethnicity Terminology

For clarity, explicitly state early in the Methods that Hmong is an ethnic group, to ensure international readers understand this categorization.

Response:

Thank you for the suggestion. The Methods now explicitly clarify ethnicity terminology. Specifically, we added a sentence in Line 120-121 stating that “Ethnicity was self-identified; response options were Lao, Hmong, Khmer, and Other. For clarity, ‘Hmong’ refers to the Hmong ethnic group”

2. Interpretation of Ethnicity vs. Socioeconomic and Geographical Factors

The discussion attributes lower tetanus seropositivity to “Hmong ethnicity.” However, ethnicity may act as a proxy for geographic remoteness, structural exclusion, or unmeasured socioeconomic barriers (e.g., lower access to health services, transportation, or education). The authors note that differences in caregiver education, occupation, and income were not statistically significant, yet these contextual factors likely influence vaccination uptake. Consider reframing to emphasize that “ethnicity” may reflect underlying structural determinants and avoid implying a direct causal effect.

Response:

Thank you for this important point. The manuscript has been revised to avoid implying a direct causal effect of ethnicity. The “Discussion” in Line 261-265 now emphasizes that the association observed for Hmong identity may reflect underlying structural determinants (e.g., geographic remoteness, transport and service availability) and sociocultural factors (e.g., language/communication barriers, trust in public services) that were not directly measured. A supporting citation from rural Lao PDR on trust-building and tailored communication improving uptake has been added (Phrasisombath et al., 2024, BMJ Global Health).

3. Association Between Vaccination and Seropositivity

Clarify that the lack of a significant association between vaccination history and diphtheria seropositivity likely reflects the persistent circulation of Diphtheria and frequent outbreaks affecting both vaccinated and unvaccinated individuals. This context is key to understanding why vaccination status was only associated with Tetanus seropositivity

Response:

Thank you for the comment. The “Discussion” in Line 249- 253 has been revised to clarify why vaccination history was associated with tetanus but not diphtheria. We added the following sentences

“In such an epidemiologic context, naturally boosted antibodies in both vaccinated and unvaccinated individuals can attenuate the association between reported vaccination history and diphtheria seropositivity. In contrast, tetanus—where subclinical infection is rare and does not typically confer protective immunity [30]—remains more closely aligned with vaccination history; observed patterns likely reflect undocumented participation in Supplementary Immunization Activities (SIAs) or maternal immunization programs [11,12].”

These edits explicitly attribute the weak diphtheria association to ongoing circulation/outbreaks and natural boosting, and explain the stronger tetanus association by the lack of natural boosting and potential contributions from SIAs/maternal immunization.

4. Sample Size Description

There is some ambiguity in the description of the sample size calculation and selection. The Methods state that “192 participants per group were selected, totaling 800,” yet also mention that “to support multistage cluster sampling, a target of approximately 960 participants was set.” This appears inconsistent and could confuse readers about how the final sample size was determined.

Please clarify:

- Precisely how the figure of 192 participants per age group was calculated (e.g., based on assumed prevalence, confidence level, design effect, expected response rate);

- why this initial total (192 × 4 age groups = 768 or “800” as written) differs from the final target of 960;

- and how this target was operationalized within the multistage cluster design (e.g., number of clusters/villages, participants per cluster, and any oversampling to account for non-response).

Providing a concise flow (calculation → adjustment for cluster design → final enrolled number) would help readers understand the sampling framework and ensure transparency of the study design.

Response:

Thank you for pointing out the ambiguity. The “Methods” in Line 96 -113 have been revised to show a clear flow from calculation → cluster design → final enrolled number:

1. How 192 per age group was calculated:

Based on prior research experience in Laos [16,17], a design effect (sample variance/population variance) of 1.6 was assumed to calculate the sample size. As the seroprevalence of diphtheria and tetanus antibodies in the general pediatric population is unknown, an estimated prevalence of 50% (p = 0.5) was used, with a 95% confidence level (Z = 1.96), a 10% margin of error (d = 0.1), and an expected response rate of 80% (R = 0.8). These parameters yielded a target of 192 participants per age group

2. Why 768 differs from the field target of 960:

Across four age groups, the minimum total under SRS assumptions is 192 multiplied by 4 equal to 768. The earlier “800” was a wording error and has been corrected. To implement a multistage cluster design with fixed integer quotas and to ensure 192 per stratum after potential non-response, the operational field target was set to 960.

3. How this target was operationalized in the cluster design:

24 villages (6 per district in four districts) were selected by probability proportional to size (PPS). Within each village, 10 participants per age group were randomly selected from household lists (40 per village), yielding 24 multiplied by 40 equal to 960 as the field target. Actual enrollment was 960 (240 per age group; 480 per province).

5. Representativeness of Selected Provinces

Please clarify how the two selected provinces (Oudomxay and Xaisomboun) compare to national averages in terms of immunization coverage, demographics, and health service access. This would help readers assess the external validity and generalizability of the findings, given that the provinces were purposively chosen to contrast coverage levels

Response:

Thank you for this comment. To aid assessment of external validity, the “Methods” now include S1 Table “The trend of Penta3 vaccination coverage over the past 10 years in each target district”, and a brief contextual paragraph in Line 88-93, comparing the two study provinces with the national context and clarifying the sampling intent:

“To situate external validity, Oudomxay generally exhibits routine immunization coverage above the national average, relatively better road connectivity, and a more mixed ethnic composition. By contrast, Xaisomboun has coverage below the national average, is mountainous, includes a higher proportion of Hmong communities, and faces greater access constraints. The two provinces were purposively selected to bracket high-coverage and low-coverage/remote contexts rather than to represent the national mean.”

This addition clarifies how the provinces compare to national patterns in coverage, demographics, and access, and explains that their purposive selection was intended to bound typical high- and low-coverage contexts.

6. Typographical Errors

- In the Statistical Analysis section, correct “seropositivity rare” to “seropositivity rate.”

Response:

Thank you very much for indicating this typographical error. I corrected this on revised manuscript.

Reviewer 2

1. Please provide the response rate of the study in both the abstract and results sections. In addition, please discuss the potential implications of selection bias arising from non-response.

Response:

Thank you for the comment. The response rate was 100% (960/960 sampled participants). This has been added to both the “Abstract” in Line 38 and the “Results” in Line 178. In the “Limitations” in Line 318-320, we note that non-response selection bias is unlikely given the 100% response; however, potential coverage/selection bias may still arise from the sampling frame (e.g., household list completeness) and the purposive selection of provinces, which may limit external validity.

2. In Table 2, please provide 95% confidence intervals (CIs) alongside the adjusted odds ratios (aORs)

Response:

Thank you for the suggestion. We revised Table 2 as follows:

• The effect-size column now reads “aOR (95% CI)” and reports 95% confidence intervals for all adjusted ORs

3. I recommend adding a graphical representation of the U-shaped seropositivity pattern by age group for each antigen (diphtheria and tetanus) stratified by province.

Response:

Thank you for your valuable suggestion. In response, we have revised the figure to highlight the U-shaped seropositivity patte

---

## [Decision Letter · Decision Letter 1]

10 Dec 2025

Seroprevalence of Diphtheria and Tetanus Antibodies among Children and Adolescents in High- and Low-Immunization Coverage Areas in the Lao People’s Democratic Republic

PONE-D-25-45990R1

Dear Dr. Yokobori,

We’re pleased to inform you that your manuscript has been judged scientifically suitable for publication and will be formally accepted for publication once it meets all outstanding technical requirements.

Kind regards,

Ray Borrow, Ph.D., FRCPath

Academic Editor

PLOS One

Additional Editor Comments (optional):

Reviewers' comments:

Reviewer's Responses to Questions

**Comments to the Author**

Reviewer #2: All comments have been addressed

2. Is the manuscript technically sound, and do the data support the conclusions?

Reviewer #2: Yes

3. Has the statistical analysis been performed appropriately and rigorously?

Reviewer #2: Yes

4. Have the authors made all data underlying the findings in their manuscript fully available?

Reviewer #2: Yes

5. Is the manuscript presented in an intelligible fashion and written in standard English?

Reviewer #2: Yes

Reviewer #2: All my concerns have been properly responded. I now suggest the editorial acceptance of the manuscript, in principle.

**Do you want your identity to be public for this peer review?** For information about this choice, including consent withdrawal, please see our Privacy Policy

Reviewer #2: No

---

## [Editor Report · Acceptance letter]

PONE-D-25-45990R1

PLOS One

Dear Dr. Yokobori,

I'm pleased to inform you that your manuscript has been deemed suitable for publication in PLOS One. Congratulations! Your manuscript is now being handed over to our production team.

Kind regards,

on behalf of

Prof. Ray Borrow

Academic Editor

PLOS One